# Sparse Regularized Optimal Transport with Deformed *q*-Entropy

**DOI:** 10.3390/e24111634

**Published:** 2022-11-10

**Authors:** Han Bao, Shinsaku Sakaue

**Affiliations:** 1Graduate School of Informatics and The Hakubi Center for Advanced Research, Kyoto University, Kyoto 604-8103, Japan; 2Department of Mathematical Informatics, Graduate School of Information Science and Technology, The University of Tokyo, Tokyo 153-8505, Japan

**Keywords:** optimal transport, Sinkhorn algorithm, convex analysis, entropy, quasi-Newton method

## Abstract

Optimal transport is a mathematical tool that has been a widely used to measure the distance between two probability distributions. To mitigate the cubic computational complexity of the vanilla formulation of the optimal transport problem, regularized optimal transport has received attention in recent years, which is a convex program to minimize the linear transport cost with an added convex regularizer. Sinkhorn optimal transport is the most prominent one regularized with negative Shannon entropy, leading to densely supported solutions, which are often undesirable in light of the interpretability of transport plans. In this paper, we report that a deformed entropy designed by *q-algebra*, a popular generalization of the standard algebra studied in Tsallis statistical mechanics, makes optimal transport solutions supported sparsely. This entropy with a deformation parameter *q* interpolates the negative Shannon entropy (q=1) and the squared 2-norm (q=0), and the solution becomes more sparse as *q* tends to zero. Our theoretical analysis reveals that a larger *q* leads to a faster convergence when optimized with the Broyden–Fletcher–Goldfarb–Shanno (BFGS) algorithm. In summary, the deformation induces a trade-off between the sparsity and convergence speed.

## 1. Introduction

Optimal transport (OT) is a classic problem in operations research, and it is used to compute a transport plan between suppliers and demanders with a minimum transportation cost. The minimum transportation cost can be interpreted as the closeness between the distributions when considering suppliers and demanders as two probability distributions. The OT problem has been extensively studied (also as the Wasserstein distance) [1] and used in robust machine learning [2], domain adaptation [3], generative modeling [4], and natural language processing [5], attributed to its many useful properties, such as the distance between two probability distributions. Recently, the OT problem has been employed for various modern applications, such as interpretable word alignment [6] and the locality-aware evaluation of object detection [7], because it can capture the geometry of data and provide a measurement method for closeness and alignment among different objects. From a computational perspective, a naïve approach is to use a network simplex algorithm or interior point method to solve the OT problem as a usual linear program; this approach requires supercubic time complexity [8] and is not scalable. A number of approaches have been suggested to accelerate the computation of the OT problem: entropic regularization [9,10], accelerated gradient descent [11], and approximation with tree [12] and graph metrics [13]. We focused our attention on entropic-regularized OT because it allows a unique solution attributed to strong convexity and transforms the original constrained optimization into an unconstrained problem with a clear primal–dual relationship. The celebrated Sinkhorn algorithm solves entropic-regularized OT with square-time complexity [9]. Furthermore, the Sinkhorn algorithm is amenable to differentiable programming, and it is easily incorporated into end-to-end learning pipelines [14,15].

Despite the popularity of the Sinkhorn algorithm, one of the main drawback is that Shannon entropy blurs the OT solution, i.e., solutions of entropic-regularized OT are always densely supported. The Shannon entropy induces a probability distribution that has strictly positive values everywhere on its support owing to the nature of the Shannon entropy [16] whereas the vanilla (unregularized) OT produces extremely sparse transport plans located on the boundaries of a polytope [17,18]. If we are interested in alignment and matching between different objects (such as in the several applications of natural language processing [6,19]), dense transport plans are not so interpretable that matching information between objects may be obfuscated by unimportant small densities contained in the transport plans. One attempt toward realizing sparse OT is to use the squared two-norm as an alternative regularizer. Blondel et al. [20] showed that the dual of this optimization problem can be solved via the L-BFGS method [21]; the primal solution corresponds to a transport plan recovered from the dual solution in a closed form, which is sparse. Although they successfully obtained a sparse OT formulation with a numerically stable algorithm, the degree of the sparsity cannot be easily modulated when we prefer to control the sparsity given a final application. Furthermore, the theoretical convergence rates of solving regularized OT are yet to be known.

In this study, we aimed to examine the relationship between the sparsity of transport plans and the convergence guarantee of regularized OT. Specifically, we propose yet another entropic regularizer called *deformed q-entropy* with a deformation parameter *q* that allows us to control the solution sparsity. We start with a dual solution of the entropic-regularized OT given by the Gibbs kernel to introduce a new regularizer; the Gibbs kernel associated with Shannon entropy induces nonsparsity, and, therefore, we replace the Gibbs kernel with another sparse kernel based on *q-exponential distribution* [22], following the idea of Tsallis statistics [23]. The deformed *q* entropy is derived from the dual solution characterized by the sparse kernel. Interestingly, the deformed *q* entropy recovers the Shannon entropy at the limit of q↗1 and matches the (negative) squared two-norm at q=0; this means that the deformed *q* entropy interpolates between the two regularizers. We confirm that the solution becomes increasingly sparse as *q* approaches zero. We call the regularized OT with the deformed *q* entropy *deformed q-optimal transport* (*q*-DOT). The *q*-DOT reveals an interesting connection between the OT solution and the *q*-exponential distribution, which is an independent interest. From the optimization perspective, we can solve the unconstrained dual of *q*-DOT with many standard solvers, as reported in Blondel et al. [20]. We can see that the convergence becomes faster with the BFGS method [24] as the deformation parameter *q* approaches one, as a result of our analysis of the convergence rate of the dual optimization. Therefore, the weaker deformation (larger *q*) leads to faster convergence while sacrificing sparsity. Finally, we demonstrate the trade-off between sparsity and convergence in the numerical experiments.

Our contributions can be summarized as: (i) showing a clear connection between the regularized OT problem and the *q*-exponential distribution; (ii) demonstrating the trade-off of the *q*-DOT between sparsity and convergence; (iii) providing a formal convergence guarantee of the *q*-DOT when solved with the BFGS method. The rest of this paper is organized as follows: Section 2 introduces the necessary background to the OT problem and entropic regularization. In Section 3, the Lagrange dual of the entropic-regularized OT problem is first shown; then, the dual optimal formula and the *q*-exponential distribution is connected to sparsify the transport matrix. Section 4 specifically focuses on the optimization perspective of the regularized OT problem, and a convergence guarantee with the BFGS method is provided, which shows the theoretical trade-off between sparsity and convergence. Finally, the empirical behavior and the trade-off of the regularized OT are numerically confirmed in Section 5.

## 2. Background

### 2.1. Preliminaries

For x∈R, let ${\left[x\right]}_{+}=x$ if x>0 and 0 otherwise, and let ${\left[x\right]}_{+}^{p}$ represent ${\left({\left[x\right]}_{+}\right)}^{p}$ hereafter. For a convex function f,X→R, where X represents a Euclidean vector space equipped with an inner product $\langle \;\cdot \;,\;\cdot \;\rangle$, the *Fenchel–Legendre conjugate* $f^{⋆}:X\to R$ is defined as $f^{⋆}\left(y\right)\sup_{x\in X}\langle x,y\rangle -f\left(x\right)$. The relative interior of a set *S* is denoted by riS, and the effective domain of a function *f* is denoted by dom(f). A differentiable function *f* is said to be *M-strongly convex* over S⊆ridom(f) if, for all x,y∈S, we have $f\left(x\right)-f\left(y\right)\leq \langle \nabla f(x),x-y\rangle -\frac{M}{2}{∥x-y∥}_{2}^{2}$. If *f* is twice differentiable, the strong convexity is equivalent to $\nabla^{2}f\left(x\right)⪰MI$ for all x∈S. Similarly, a differentiable function *f* is said to be *M-smooth* over S⊆ridom(f) if for all x,y∈S, we have ${∥\nabla f\left(x\right)-\nabla f\left(y\right)∥}_{2}\leq M{∥x-y∥}_{2}$, which is equivalent to $\nabla^{2}f\left(x\right)⪯MI$ for all x∈S if *f* is twice differentiable.

### 2.2. Optimal Transport

The OT is a mathematical problem to find a transport plan between two probability distributions with the minimum transport cost. The discussions in this paper are restricted to discrete distributions. Let (X,d), $\delta_{x}$, and ${▵}^{n-1}:=\left\{p\in {\left[0,1\right]}^{n}|\langle p,1_{n}\rangle =1\right\}$ represent a metric space, Dirac measure at point x, and (n−1)-dimensional probability simplex, respectively. Let $\mu =\sum_{i=1}^{n}a_{i}\delta_{x_{i}}$ and $\nu =\sum_{j=1}^{m}b_{i}\delta_{y_{i}}$ be histograms supported on the finite sets of points ${\left(x_{i}\right)}_{i=1}^{n}\subseteq X$ and ${\left(y_{j}\right)}_{j=1}^{m}\subseteq X$, respectively, where $a\in {▵}^{n-1}$ and $b\in {▵}^{m-1}$ are probability vectors. The OT between two discrete probability measures μ and ν is the optimization problem
(1)$$T\left(\mu ,\nu \right):=\underset{\Pi \in U(\mu ,\nu )}{\inf}\sum_{i=1}^{n}\sum_{j=1}^{m}d\left(x_{i},y_{j}\right)\Pi_{ij},$$
where U represents the transport polytope, defined as
(2)$$U\left(\mu ,\nu \right):=\left\{\Pi \in R_{\geq 0}^{n\times m}|\Pi 1_{m}=a,\Pi^{⊤}1_{n}=b\right\}.$$
The transport polytope U defines the constraints on the row/column marginals of a transport matrix Π. These constraints are often referred to as coupling constraints. For notational simplicity, matrix $D_{ij}:=d\left(x_{i},y_{j}\right)$ and expectation $\langle D,\Pi \rangle :=\sum_{i=1}^{n}\sum_{j=1}^{m}D_{ij}\Pi_{ij}$ are used hereafter. T(μ,ν) is known as a *1-Wasserstein distance*, which defines a metric space over histograms [1].

Equation (Equation 1) is a linear program and can be solved by well-studied algorithms such as the interior point and network simplex methods. However, its computational complexity is $O(n^{3}\log n)$ (assuming n=m), so is not scalable to large datasets [8].

### 2.3. Entropic Regularization and Sinkhorn Algorithm

The entropic-regularized formulation is commonly used to reduce the computational burden. Here, we introduce regularized OT with negative Shannon entropy [9] as
(3)$$T_{-\lambda H}\left(\mu ,\nu \right):=\underset{\Pi \in U(\mu ,\nu )}{\inf}\langle D,\Pi \rangle +\lambda \underset{\mathrm{negative}\;\mathrm{Shannon}\;\mathrm{entropy}}{\underset{︸}{\sum_{i=1}^{n}\sum_{j=1}^{m}\left(\Pi_{ij}\log\Pi_{ij}-\Pi_{ij}\right)}},$$
where λ>0 represents the regularization strength. Let us review the derivation of the updates of the Sinkhorn algorithm. The Lagrangian of the optimization problem in Equation (Equation 3) is
(4)$$\begin{array}{cc} L(\Pi ,\alpha ,\beta ):= & \sum_{i=1}^{n}\sum_{j=1}^{m}\left(D_{ij}\Pi_{ij}+\lambda \left(\Pi_{ij}\log\Pi_{ij}-\Pi_{ij}\right)\right)\\ \; & \;+\sum_{i=1}^{n}\alpha_{i}\left({\left[\Pi 1_{m}\right]}_{i}-a_{i}\right)+\sum_{j=1}^{m}\beta_{j}\left({\left[\Pi^{⊤}1_{n}\right]}_{j}-b_{j}\right), \end{array}$$
where $\alpha \in R^{n}$ and $\beta \in R^{m}$ represent the Lagrangian multipliers. Equation (Equation 4) ignores the constraints $\Pi_{ij}\geq 0$ (for all i∈[n] and j∈[m]); however, they will be automatically satisfied. By taking the derivative in $\Pi_{ij}$,
(5)$$\nabla_{\Pi_{ij}}L=D_{ij}+\lambda \log\Pi_{ij}+\alpha_{i}+\beta_{j},$$
and, hence, the stationary condition $\nabla_{\Pi_{ij}}L=0$ induces the solution
(6)$$\Pi_{ij}=\exp\left(-\frac{\alpha_{i}+\beta_{j}+D_{ij}}{\lambda }\right).$$
The decomposition $\Pi_{ij}=\exp\left(-\frac{D_{ij}}{\lambda }\right)/\exp\left(\frac{\alpha_{i}+\beta_{j}}{\lambda }\right)$ suggests that the stationary point is the (normalized) *Gibbs kernel* $\exp\left(-\frac{D_{ij}}{\lambda }\right)$. One can easily infer that the Sinkhorn solution is dense because the Gibbs kernel is supported on the entire $R_{\geq 0}$, i.e., $\exp\left(-\frac{z}{\lambda }\right)>0$ for all $z\in R_{\geq 0}$. We can write Equation (Equation 6) into a matrix form by applying the variable transforms $u_{i}:=\exp\left(-\frac{\alpha_{i}}{\lambda }\right)$, $v_{j}:=\exp\left(-\frac{\beta_{j}}{\lambda }\right)$, and $K_{ij}:=\exp\left(-\frac{D_{ij}}{\lambda }\right)$ as
(7)$$\Pi =\underset{:=U}{\underset{︸}{\mathrm{diag}(u)}}\;K\underset{:=V}{\underset{︸}{\mathrm{diag}(v)}}.$$
The following Sinkhorn updates are used to make Equation (Equation 7) meet the marginal constraints:(8)$$\left\{\begin{array}{c} u^{\prime }\leftarrow a/\left(Kv\right)\\ v^{\prime }\leftarrow b/\left(K^{⊤}u\right) \end{array}\right.,$$
where z/η represents the element-wise division of the two vectors z and η. The computational complexity is O(Knm) because the Sinkhorn updates involve only matrix-vector multiplications and element-wise divisions; *K* represents the number of the Sinkhorn updates. Finer analysis of the number of updates required to meet the error tolerance is provided in the literature [25].

## 3. Deformed *q*-Entropy and *q*-Regularized Optimal Transport

### 3.1. Regularized Optimal Transport and Its Dual

Let us consider the following primal problem with a general regularization function Ω.

**Definition** **1**(Primal of regularized OT).
(9)$$T_{\Omega }\left(\mu ,\nu \right)=\underset{\Pi \in U(\mu ,\nu )}{\inf}\langle D,\Pi \rangle +\sum_{i,j}\Omega \left(\Pi_{ij}\right),$$
*where Ω:R→R represents a proper closed convex function.*

Next, we derive its dual by Lagrange duality. The Lagrangian of Equation (Equation 9) is defined as
(10)$$L\left(\Pi ,\alpha ,\beta \right):=\langle D,\Pi \rangle +\sum_{i,j}\Omega \left(\Pi_{ij}\right)+\langle \alpha ,\Pi 1_{m}-a\rangle +\langle \beta ,\Pi^{⊤}1_{n}-b\rangle ,$$
with dual variables $\alpha \in R^{n}$ and $\beta \in R^{m}$. Then, the primal can be rewritten in terms of the Lagrangian
(11)$$T_{\Omega }\left(\mu ,\nu \right)=\underset{\Pi \in R_{\geq 0}^{n\times m}}{\inf}\underset{\alpha \in R^{n},\beta \in R^{m}}{\sup}L\left(\Pi ,\alpha ,\beta \right).$$
In this Lagrangian formulation, we let the constraints $\Pi \in R_{\geq 0}^{n\times m}$ remain for a technical reason. The constrained optimization problem in (Equation 11) can be reformulated into the following unconstrained one with an indicator function $I_{R_{\geq 0}^{n\times m}}$.
(12)$$T_{\Omega }\left(\mu ,\nu \right)=\underset{\Pi \in R^{m\times m}}{\inf}\underset{\alpha \in R^{n},\beta \in R^{m}}{\sup}L\left(\Pi ,\alpha ,\beta \right)+I_{R_{\geq 0}^{n\times m}}\left(\Pi \right),$$
which corresponds to an optimization problem with the convex objective function $\langle D,\Pi \rangle +\sum_{i,j}\Omega \left(\Pi_{ij}\right)+I_{R_{\geq 0}^{n\times m}}\left(\Pi \right)$ with only the linear constraints $\Pi 1_{m}=a$ and $\Pi^{⊤}1_{n}=b$. By invoking the Sinkhorn–Knopp theorem [26], the existence of a strictly feasible solution, namely, a solution satisfying $\Pi 1_{m}=a$ and $\Pi^{⊤}1_{n}=b$, can be confirmed. Hence, we see that the Slater condition is satisfied, and the strong duality holds as follows:(13)$$\begin{array}{cc} T_{\Omega } & \left(\mu ,\nu \right)=\underset{\alpha \in R^{n},\beta \in R^{m}}{\sup}\underset{\Pi \in R_{\geq 0}^{n\times m}}{\inf}L\left(\Pi ,\alpha ,\beta \right)\\ \; & =\underset{\alpha \in R^{n},\beta \in R^{m}}{\sup}-\langle a,\alpha \rangle -\langle b,\beta \rangle +\underset{\Pi \in R_{\geq 0}^{n\times m}}{\inf}\sum_{i,j}\left(D_{ij}+\alpha_{i}+\beta_{j}\right)\Pi_{ij}+\Omega \left(\Pi_{ij}\right)\\ \; & =\underset{\alpha \in R^{n},\beta \in R^{m}}{\sup}-\langle a,\alpha \rangle -\langle b,\beta \rangle -\left(\underset{\Pi \in R_{\geq 0}^{n\times m}}{\sup}\sum_{i,j}-\left(D_{ij}+\alpha_{i}+\beta_{j}\right)\Pi_{ij}-\Omega \left(\Pi_{ij}\right)\right)\\ \; & =\underset{\alpha \in R^{n},\beta \in R^{m}}{\sup}-\langle a,\alpha \rangle -\langle b,\beta \rangle -\sum_{i,j}\Omega^{⋆}\left(-D_{ij}-\alpha_{i}-\beta_{j}\right), \end{array}$$
where $\Omega^{⋆}$ represents the Fenchel–Legendre conjugate of Ω:R→R
(14)$$\Omega^{⋆}\left(\eta \right):=\underset{\pi \geq 0}{\sup}\eta \pi -\Omega \left(\pi \right).$$
Although each element of the transport plans ranges over [0,1], it is sufficient to define the Fenchel–Legendre conjugate as the supremum over $R_{\geq 0}$ because of how $\Omega^{⋆}$ emerges in the strong duality (Equation 13). According to Danskin’s theorem [27], the supremum of the Fenchel–Legendre conjugate can be attained at
(15)$$\Pi_{ij}^{⋆}=\nabla \Omega^{⋆}\left(-D_{ij}-\alpha_{i}-\beta_{j}\right).$$
Therefore, the dual of regularized OT is formulated as follows:

**Definition** ** 2**(Dual of regularized OT).
(16)$$T_{\Omega }\left(\mu ,\nu \right)=\underset{\alpha \in R^{n},\beta \in R^{m}}{\sup}-\langle a,\alpha \rangle -\langle b,\beta \rangle -\sum_{i,j}\Omega^{⋆}\left(-D_{ij}-\alpha_{i}-\beta_{j}\right),$$
*where $\Omega^{⋆}$ represents the Fenchel–Legendre conjugate $\Omega^{⋆}\left(\eta \right):=\sup_{\pi \geq 0}\eta \pi -\Omega \left(\pi \right)$. The optimal solution of the primal is given by the dual map $\nabla \Omega^{⋆}$ such that $\Pi_{ij}^{⋆}=\nabla \Omega^{⋆}\left(-D_{ij}-\alpha_{i}^{⋆}-\beta_{j}^{⋆}\right)$, where $\left(\alpha^{⋆},\beta^{⋆}\right)$ represents the dual optimal solution.*

Next, we see several examples that are summarized in Table 1.

**Example** **1**(Negative Shannon entropy). *Let Ω(π)=−λH(π)=λ(πlogπ−π); then $\Omega^{⋆}\left(\eta \right)=\lambda e^{\eta /\lambda }$ and $\nabla \Omega^{⋆}\left(\eta \right)=e^{\eta /\lambda }$. The optimal solution represented with the optimal dual variables $\left(\alpha^{⋆},\beta^{⋆}\right)$ is $\Pi_{ij}^{⋆}=\exp\left(-\frac{D_{ij}+\alpha_{i}^{⋆}+\beta_{j}^{⋆}}{\lambda }\right)$. This recovers the stationary point of the Sinkhorn OT in Equation (Equation 6). The solution is dense because the regularizer* Ω *induces the Gibbs kernel $\nabla \Omega^{⋆}\left(\eta \right)=e^{\eta /\lambda }>0$ for all η∈R.*

**Example** **2**(Squared 2-norm). *Let $\Omega \left(\pi \right)=\frac{\lambda }{2}\pi^{2}$; then $\Omega^{⋆}\left(\eta \right)=\frac{1}{2\lambda }{\left[\eta \right]}_{+}^{2}$ and $\nabla \Omega^{⋆}\left(\eta \right)=\frac{1}{\lambda }{\left[\eta \right]}_{+}$. The optimal solution represented with the optimal dual variables $\left(\alpha^{⋆},\beta^{⋆}\right)$ is $\Pi_{ij}^{⋆}=\frac{1}{\lambda }{\left[-D_{ij}-\alpha_{i}^{⋆}-\beta_{j}^{⋆}\right]}_{+}$. As mentioned by Blondel et al. [20], the squared 2-norm can sparsify the solution because $\nabla \Omega^{⋆}\left(\eta \right)=\frac{1}{\lambda }{\left[\eta \right]}_{+}$ may take the value 0.*

### 3.2. *q* Algebra and Deformed Entropy

As shown in the last few examples, the dual map $\nabla \Omega^{⋆}$ plays an important role in the OT solution sparsity. In addition, the induced $\nabla \Omega^{⋆}$ is the Gibbs kernel when the negative Shannon entropy is used as Ω. Therefore, one may think of designing a regularizer from $\nabla \Omega^{⋆}$ by utilizing a kernel function that induces sparsity. One candidate is a *q-exponential distribution*. We begin with some basics required to formulate *q*-exponential distributions.

First, we introduce *q-algebra*, which has been well studied in the field of Tsallis statistical mechanics [23,29,30]. *q* algebra has been used in the machine-learning literature for regression [31], Bayesian inference [32], and robust learning [33]. For a deformation parameter q∈[0,1], the *q*-logarithm and *q*-exponential functions are defined as
(17)$$\begin{array}{ccc} \log_{q}\left(x\right):=\left\{\begin{array}{cc} \frac{x^{1-q}-1}{1-q} & \mathrm{if}\;q\in [0,1)\\ \log(x) & \mathrm{if}\;q=1 \end{array}\right., & \; & \exp_{q}\left(x\right):=\left\{\begin{array}{cc} {\left[1+\left(1-q\right)x\right]}_{+}^{1/(1-q)} & \mathrm{if}\;q\in [0,1)\\ \exp(x) & \mathrm{if}\;q=1 \end{array}\right.. \end{array}$$
The *q* logarithm is defined for only x>0, as in the natural logarithm; they are inverse functions to each other (in an appropriate domain) and they recover the natural definition of the logarithm and exponential as q↗1. Their derivatives are ${\left(\log_{q}\left(x\right)\right)}^{\prime }=\frac{1}{x^{q}}$ and ${\left(\exp_{q}\left(x\right)\right)}^{\prime }=\exp_{q}{\left(x\right)}^{q}$, respectively. The additive factorization property exp(x+y)=exp(x)exp(y) satisfied by the natural exponential no longer holds for the *q* exponential, such that $\exp_{q}\left(x+y\right)\neq \exp_{q}\left(x\right)\exp_{q}\left(y\right)=\exp_{q}\left(x+y+\left(1-q\right)xy\right)$. Instead, we can construct another algebraic structure by introducing the other operation called the *q* product ${⊗}_{q}$:(18)$$x{⊗}_{q}y={\left[x^{1-q}+y^{1-q}-1\right]}_{+}^{1/(1-q)}.$$
With this product, the pseudoadditive factorization $\exp_{q}\left(x+y\right)=\exp_{q}\left(x\right){⊗}_{q}\exp_{q}\left(y\right)$ holds. Thus, the *q* algebra captures rich nonlinear structures, and it is often used to extend the Shannon entropy to the Tsallis entropy [23]
(19)$$T_{q}\left(\pi \right)=-\sum_{i=1}^{n}\pi_{i}^{q}\log_{q}\left(\pi_{i}\right).$$
One can see that the Tsallis entropy has an equivalent power formulation $T_{q}\left(\pi \right)=\sum_{i=1}^{n}\frac{\pi_{i}-\pi_{i}^{q}}{1-q}$, which means that it is often suitable for modeling heavy-tailed phenomena such as the power law. Although the introduced *q* logarithm and exponential can look arbitrary, they can be axiomatically derived by assuming the essential properties of the algebra (see Naudts [29]). For more physical insights, we recommend readers to refer to the literature [30].

Next, we introduce the *q-exponential distribution*. We introduce a simpler form for our purpose, whereas more general formulations of the *q*-exponential distribution have been introduced in the literature [22]. Given the form of the Gibbs kernel k(ξ):=exp(−ξ/λ), we define the *q-Gibbs kernel* as follows:

**Definition** ** 3**(*q*-Gibbs kernel). *For ξ≥0, we define the q-Gibbs kernel as $k_{q}\left(\xi \right)\exp_{q}\left(-\xi /\lambda \right)$ for a deformation parameter q∈[0,1] and a temperature parameter $\lambda \in R_{>0}$.*

If we take ξ as the (centered) squared distance, then $k_{q}\left(\xi \right)$ represents the *q*-Gaussian distribution [22]. We illustrate the *q*-Gibbs kernel with different deformation parameters in Figure 1.

By definition, the support of the *q*-Gibbs kernel is $\mathrm{supp}\left(k_{q}\right)=\left[0,\frac{\lambda }{1-q}\right]$ for q∈[0,1) and $\mathrm{supp}\left(k_{q}\right)=R_{\geq 0}$ for q=1. This indicates that the *q*-Gibbs kernel ignores the effect of a too-large ξ (or too large a distance between two points); its threshold is smoothly controlled by the temperature parameter λ and deformation parameter *q*.

Finally, we derive an entropic regularizer that induces sparsity by using the *q*-Gibbs kernel. Given the stationary condition in Equation (Equation 15), we impose the following functional form on the dual map:(20)$$\pi =\nabla \Omega^{⋆}\left(\eta \right)=\exp_{q}\left(\frac{\eta }{\lambda }\right),$$
where $\left(\pi ,\eta \right)=\left(\Pi_{ij}^{⋆},-D_{ij}-\alpha_{i}-\beta_{j}\right)$. Equation (Equation 20) results in the factorization
(21)$$\Pi_{ij}^{⋆}=\exp_{q}\left(-\frac{\alpha_{i}}{\lambda }\right){⊗}_{q}\exp_{q}\left(-\frac{-D_{ij}}{\lambda }\right){⊗}_{q}\exp_{q}\left(-\frac{-\beta_{j}}{\lambda }\right),$$
and a sufficiently large input distance $D_{ij}$ drives $\Pi_{ij}$ to zero; though $\exp_{q}\left(-D_{ij}/\lambda \right)=0$ does not immediately imply $\Pi_{ij}^{⋆}=0$ because the *q*-product ${⊗}_{q}$ lacks an absorbing element. By solving Equation (Equation 20),
(22)$$\nabla \Omega \left(\pi \right)=\lambda \log_{q}\left(\pi \right),\;\Omega \left(\pi \right)=\frac{\lambda }{2-q}\left(\pi \log_{q}\left(\pi \right)-\pi \right).$$
For the completeness, its derivation is shown in Appendix A. Hence, we define the *deformed q entropy* as follows:

**Definition** ** 4**(Deformed *q*-entropy). *For $\pi \in {▵}^{n-1}$, the* deformed *q* entropy *is defined as*
(23)$$H_{q}\left(\pi \right)=-\frac{1}{2-q}\sum_{i=1}^{n}\left(\pi_{i}\log_{q}\left(\pi_{i}\right)-\pi_{i}\right).$$
*The deformed q-entropic regularizer for an element $\pi_{i}$ is $\Omega \left(\pi_{i}\right)=\frac{\lambda }{2-q}\left(\pi_{i}\log_{q}\left(\pi_{i}\right)-\pi_{i}\right)$.*

The deformed *q* entropy recovers the Shannon entropy at the limit q↗1: $H_{1}\left(\pi \right)=-\sum_{i}\left(\pi_{i}\log\left(\pi_{i}\right)-\pi_{i}\right)$. In addition, the limit q↘0 recovers the negative of the squared 2-norm: $H_{0}\left(\pi \right)=-\frac{1}{2}\sum_{i}\left(\pi_{i}^{2}-2\pi_{i}\right)=-\frac{1}{2}{∥\pi ∥}_{2}^{2}+1$. Therefore, the deformed *q* entropy is an interpolation between the Shannon entropy and squared 2-norm. Hereafter, we consider the regularized OT with the deformed *q* entropy
(24)$$T_{-\lambda H_{q}}\left(\mu ,\nu \right)=\underset{\Pi \in U(\mu ,\nu )}{\inf}\langle D,\Pi \rangle -\lambda H_{q}\left(\Pi \right),$$
by solving its dual counterpart. The deformed *q* entropy is different from the Tsallis entropy $T_{q}$ (see Equation (Equation 19)) in that the Tsallis entropy and deformed *q* entropy are defined by the *q expectation*$\langle \pi^{q},\;\cdot \;\rangle$ [34] and the usual expectation $\langle \pi ,\;\cdot \;\rangle$, respectively, while both are defined by the *q* logarithm.

**Remark** **1.**
*The primary reason we picked the deformed q entropy $H_{q}$ to design the regularizer is owing to its natural connection to the q-Gibbs kernel through the dual map, $\nabla {\left(-\lambda H_{q}\right)}^{⋆}\left(\eta \right)=\exp_{q}\left(\eta /\lambda \right)$. When the Tsallis entropy $T_{q}$ is used, the dual map is*

(25)
$$\nabla {\left(-\lambda T_{q}\right)}^{⋆}\left(\eta \right)=\frac{q^{1/(1-q)}}{\exp_{q}\left(-\eta /\lambda \right)},$$

*which is not naturally connected to the q-Gibbs kernel.  Muzellec et al. [35] proposed regularized OT with the Tsallis entropy, but they did not discuss its sparsity. As we show in Section D.1, the Tsallis entropy does not empirically induce sparsity.*


In Figure 2, the deformed *q* entropy with a different deformation parameter is plotted for the one-dimensional simplex ${▵}^{1}$. One can easily confirm that $H_{q}\left(\pi \right)$ is concave for $\pi \in R_{\geq 0}^{n}$, as illustrated in the figure.

## 4. Optimization and Convergence Analysis

### 4.1. Optimization Algorithm

We occasionally write $\Omega =-\lambda H_{q}$ to simplify the notation in this section. By simple algebra, we confirm
(26)$$\Omega^{⋆}\left(\eta \right)=\frac{\lambda }{2-q}\exp_{q}{\left(\frac{\eta }{\lambda }\right)}^{2-q},$$
which is convex because of the concavity of $H_{q}$. To solve Equation (Equation 24), we solve the dual
(27)$$T_{-\lambda H_{q}}\left(\mu ,\nu \right)=\underset{\alpha \in R^{n},\beta \in R^{m}}{\sup}\underset{:=-F(z)}{\underset{︸}{-\langle a,\alpha \rangle -\langle b,\beta \rangle -\frac{\lambda }{2-q}\sum_{i,j}\exp_{q}{\left(-\frac{D_{ij}+\alpha_{i}+\beta_{j}}{\lambda }\right)}^{2-q}}},$$
where z:=(α,β) denotes dual variables. As Equation (Equation 27) is an unconstrained optimization problem, many famous optimization solvers can be used to solve it; here, we use the BFGS method [24]. For the sake of convergence analysis (Section 4.2), we optimize the convex $\ell_{2}$-regularized dual objective
(28)$$\begin{array}{cc} \mathrm{minimize}\; & \tilde{F}\left(z\right):=\langle a,\alpha \rangle +\langle b,\beta \rangle +\sum_{i,j}\Omega^{⋆}\left(-D_{ij}-\alpha_{i}-\beta_{j}\right)+\frac{\kappa }{2}{∥z∥}_{2}^{2}, \end{array}$$
where κ>0 represents the $\ell_{2}$-regularization parameter. In practice, $\ell_{2}$ regularization hardly affects the performance when κ is sufficiently small. We can characterize the convergence rate by introducing (small) $\ell_{2}$ regularization, which makes the objective strongly convex, whereas the convergence guarantee without its rate is still possible without $\ell_{2}$ regularization [36].

We briefly summarize the algorithm in Algorithm 1, where $d^{\left(k\right)}$, $\rho^{\left(k\right)}$, and $g^{\left(k\right)}:=\nabla \tilde{F}\left(z^{\left(k\right)}\right)$ represent the *k*th update direction, *k*th step size, and gradient at the current variable $z^{\left(k\right)}$, respectively.
(29)$$s^{\left(k\right)}:=z^{\left(k+1\right)}-z^{\left(k\right)}\;\mathrm{and}\;\zeta^{\left(k\right)}:=g^{\left(k+1\right)}-g^{\left(k\right)}$$
are the differences of the dual variables and gradients between the next and current steps, respectively. Furthermore, let $\left(\gamma ,\gamma^{\prime }\right)$ be the tolerance parameter for the Wolfe conditions, i.e., update directions and step sizes satisfy the conditions
(30)$$\begin{array}{cccc} \tilde{F}\left(z^{\left(k\right)}+\rho^{\left(k\right)}d^{\left(k\right)}\right) & \leq \tilde{F}\left(z^{\left(k\right)}\right)+\gamma^{\prime }\rho^{\left(k\right)}g^{(k)⊤}d^{\left(k\right)}, & \; & (\mathrm{Armijo}\;\mathrm{condition}) \end{array}$$
(31)$$\begin{array}{cccc} g^{(k+1)⊤}d^{\left(k\right)} & \geq \gamma g^{(k)⊤}d^{\left(k\right)}. & \; & (\mathrm{curvature}\;\mathrm{condition}) \end{array}$$

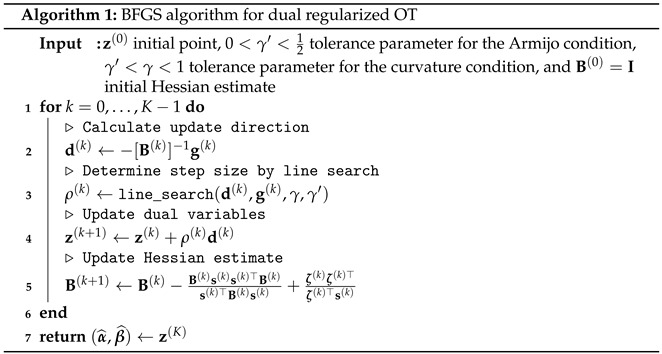


After obtaining the dual solution $\left(\hat{\alpha },\hat{\beta }\right)$, the primal solution can be recovered from Equation (Equation 15).

### 4.2. Convergence Analysis

We provide a convergence guarantee for Algorithm 1. A technical assumption is stated beforehand.

**Assumption** **1.**
*Let $z_{⋆}$ be the global optimum of $\tilde{F}$. For τ∈(0,1), we define the set $Z_{\tau }\subseteq \mathrm{ri}\mathrm{dom}\left(\tilde{F}\right)$ as*

(32)
$$Z_{\tau }:=\left\{z|\nabla \Omega^{⋆}\left(-D_{ij}-\alpha_{i}-\beta_{j}\right)\leq \tau \;\mathrm{for}\;\mathrm{all}\;i,j\right\}.$$

*Assume that $z^{\left(K\right)}$ obtained by Algorithm 1 and $z_{⋆}$ are contained in $Z_{\tau }$.*


The dual map $\nabla \Omega^{⋆}$ translates dual variables into primal variables, as in Equation (Equation 15). It is easy to confirm that $Z_{\tau }$ is a closed convex set attributed to the convexity of $\nabla \Omega^{⋆}$. Assumption 1 essentially assumes that all elements of the primal matrix (of $z^{\left(K\right)}$ and $z_{⋆}$) are strictly less than 1; this always holds for $z_{⋆}$ (unless n=m=1) because of the strong duality. Moreover, this assumption is natural for $z^{\left(K\right)}$ values sufficiently close to the optimum $z_{⋆}$. The bound parameter τ is a key element for characterizing the convergence speed.

**Theorem** **1.**
*Let N:=max{n,m}. Under Assumption 1, Algorithm 1 with the parameter choice $\kappa =2N\tau^{q}\lambda^{-1}$ returns a point $z^{\left(k\right)}$ satisfying*

(33)
$$∥g^{\left(K\right)}{∥}_{2}<\sqrt{\frac{16\left(\tilde{F}\left(z^{\left(0\right)}\right)-{\tilde{F}}_{⋆}\right)N\tau^{q}}{\lambda }}r^{K}$$

*where ${\tilde{F}}_{⋆}:=\inf_{z}\tilde{F}\left(z\right)$ represents the optimal value of the $\ell_{2}$-regularized dual objective and 0<r<1 is an absolute constant independent from (λ,τ,q,N).*


The proof is shown in Section 4.3. We conclude that a larger deformation parameter *q* yields better convergence because the coefficient in Equation (Equation 33) is $O(\tau^{q/2})$ with the base $\sqrt{\tau }<1$. Therefore, the deformation parameter introduces a new trade-off: *q↘0 yields a more sparse solution but slows down the convergence, whereas q↗1 ameliorates the convergence while sacrificing sparsity*. One may obtain the solution faster than the squared 2-norm regularizer used in Blondel et al. [20], which corresponds to the case q=0, by modulating the deformation parameter *q*.

In regularized OT, it is a common approach to use weaker regularization (i.e., a smaller λ) to obtain a solution sparser and closer to the unregularized solution; however, a smaller λ results in numerical instability and slow computation [37]. This can be observed from Equation (Equation 33) because a smaller λ drives its upper bound considerably large.

Subsequently, we compared the computational complexity of *q*-DOT with the BFGS method and Sinkhorn algorithm. Altschuler et al. [25] showed that the Sinkhorn algorithm satisfies coupling constraints within the $\ell_{1}$ error ε in $O(N^{2}\left(\log N\right)\varepsilon^{-3})$ time, which is the sublinear convergence rate. In contrast, our convergence rate in Equation (Equation 33) is translated into the iteration complexity $K=O(\log\left(N\varepsilon^{-1}\right))$, where $∥g^{\left(K\right)}{∥}_{2}\leq \varepsilon$. The gradient of $\tilde{F}$ is
(34)$$\nabla \tilde{F}\left(z\right)=\left[\begin{array}{c} \vdots \\ a_{i}-\sum_{j=1}^{m}\nabla \Omega^{⋆}\left(-D_{ij}-\alpha_{i}-\beta_{j}\right)+\kappa \alpha_{i}\\ \vdots \\ b_{i}-\sum_{i=1}^{n}\nabla \Omega^{⋆}\left(-D_{ij}-\alpha_{i}-\beta_{j}\right)+\kappa \beta_{j}\\ \vdots \end{array}\right],$$
and $\nabla \Omega^{⋆}\left(\;\cdot \;\right)$ represents the mapping from the dual variables $\left(\alpha_{i},\beta_{j}\right)$ to the primal transport matrix $\Pi_{ij}$ in Equation (Equation 15). Therefore, the gradient norm of F and coupling constraint error are comparable when the $\ell_{2}$-regularization parameter κ is sufficiently small. The overall computational complexity is $O(N^{2}\log\left(N\varepsilon^{-1}\right))$ because the one step of Algorithm 1 runs in $O(N^{2})$ time; this is the linear convergence rate. To confirm the one step of Algorithm 1 runs in $O(N^{2})$ time, we note that the update direction can be computed with $O(N^{2})$ time by using the Sherman–Morrison formula to invert $B^{\left(k\right)}$. In addition, the Hessian estimate can be updated with $O(N^{2})$ time because $B^{\left(k\right)}$ is the rank-1 update and the computation of its inverse only requires the matrix-vector products of size *N*. Hence, Algorithm 1 exhibits better convergence in terms of the stopping criterion ε. The comparison is summarized in Table 2.

### 4.3. Proofs

To prove Theorem 1, we leveraged several lemmas shown below. Lemma 2 is based on Powell [24] and Byrd et al. [36]. The missing proofs are provided in Appendix C.

**Lemma** **1.**
*For the initial point $z^{\left(0\right)}$ and sequence $z^{\left(1\right)},z^{\left(2\right)},\ldots ,z^{\left(K\right)}$ obtained by Algorithm 1, we define the following set and its bound:*

(35)
$$Z:=\mathrm{conv}\left(\left\{z^{\left(0\right)},z^{\left(1\right)},z^{\left(2\right)},\ldots ,z^{\left(K\right)}\right\}\right),\;R:=\underset{z\in Z}{\sup}\max_{i,j}\nabla \Omega^{⋆}\left(-D_{ij}-\alpha_{i}-\beta_{j}\right),$$

*where conv(S) represents the convex hull of the set S. Then, $\tilde{F}:R^{n+m}\to R$ is $M_{1}$ strongly convex and $M_{2}$-smooth over Z, where $M_{1}=\kappa$ and $M_{2}\leq \kappa +2NR^{q}\lambda^{-1}$. Moreover, $\tilde{F}$ is $M_{2}^{\prime }$-smooth over $Z_{\tau }$ (defined in Equation (Equation 32)), where $M_{2}^{\prime }\leq \kappa +2N\tau^{q}\lambda^{-1}$.*


**Lemma** **2.**
*Let $z^{\left(1\right)},z^{\left(2\right)},\ldots ,z^{\left(k\right)}$ be a sequence generated by Algorithm 1 given an initial point $z^{\left(0\right)}$. In addition, let $c_{1}$, $c_{2}$, $c_{3}$, $c_{4}$, and $c_{5}$ be the constants*

(36)
$$\begin{array}{cccc} c_{1} & :=\frac{1-\gamma }{M_{2}}, & \; & \;c_{2}:=\frac{n+m}{K}+M_{2},\\ c_{3} & :={\left(\frac{K}{n+m}\right)}^{(n+m)/K}c_{2}^{\frac{n+m+K}{K}}, & \; & \;c_{4}:=\frac{c_{3}}{1-\gamma },\\ c_{5} & :=\frac{2(1-\gamma^{\prime })}{M_{1}}. \end{array}$$

*Then,*

(37)
$$\tilde{F}\left(z^{\left(K\right)}\right)-{\tilde{F}}_{⋆}\leq {\left(1-\frac{\gamma^{\prime }c_{1}M_{1}}{2c_{4}^{2}c_{5}^{2}}\right)}^{K/2}\left(\tilde{F}\left(z^{\left(0\right)}\right)-{\tilde{F}}_{⋆}\right).$$



**Lemma** **3.**
*Let $c_{1}$, $c_{2}$, $c_{3}$, $c_{4}$, and $c_{5}$ be the same constants defined in Lemma 2. Then,*

(38)
$$\frac{\gamma^{\prime }c_{1}M_{1}}{c_{4}^{2}c_{5}^{2}}>\frac{{\left(1-\gamma \right)}^{3}\gamma^{\prime }e^{-2(n+m)/e}}{4{\left(1-\gamma^{\prime }\right)}^{2}}{\left(\frac{M_{1}}{M_{2}}\right)}^{3}.$$



**Proof of Theorem** **1.**Because $\tilde{F}$ is differentiable and strongly convex, there exists an optimum $z_{⋆}$ such that $g_{⋆}:=\nabla \tilde{F}\left(z_{⋆}\right)=0$; this implies $∥g^{\left(K\right)}{∥}_{2}={∥g^{\left(K\right)}-g_{⋆}∥}_{2}$.By using Assumption 1 and Lemma 1, we obtain $∥g^{\left(K\right)}-g_{⋆}{∥}_{2}=∥\nabla \tilde{F}\left(z^{\left(K\right)}\right)-\nabla \tilde{F}\left(z_{⋆}\right){∥}_{2}\leq M_{2}^{\prime }{∥z^{\left(K\right)}-z_{⋆}∥}_{2}$. In addition, $∥z^{\left(K\right)}-z_{⋆}{∥}_{2}^{2}\leq \frac{2}{M_{1}}\left(\tilde{F}\left(z^{\left(K\right)}\right)-{\tilde{F}}_{⋆}\right)$ as $\tilde{F}$ is $M_{1}$ strongly convex over Z and the stationary condition $\nabla \tilde{F}\left(z_{⋆}\right)=0$ holds. We obtain the convergence bound by using Lemmas 2 and 3 as
(39)$$\begin{array}{cc} ∥g^{\left(K\right)}{∥}_{2} & =∥g^{\left(K\right)}-g_{⋆}{∥}_{2}\\ \; & \leq M_{2}^{\prime }{∥z^{\left(K\right)}-z_{⋆}∥}_{2}\\ \; & \leq M_{2}^{\prime }\sqrt{\frac{2(\tilde{F}\left(z^{\left(K\right)}\right)-{\tilde{F}}_{⋆})}{M_{1}}}\\ \; & \leq M_{2}^{\prime }\sqrt{\frac{2(\tilde{F}\left(z^{\left(0\right)}\right)-{\tilde{F}}_{⋆})}{M_{1}}{\left(1-\frac{\gamma^{\prime }c_{1}M_{1}}{2c_{4}^{2}c_{5}^{2}}\right)}^{K/2}}\\ \; & <M_{2}^{\prime }\sqrt{\frac{2(\tilde{F}\left(z^{\left(0\right)}\right)-{\tilde{F}}_{⋆})}{M_{1}}{\left(1-\frac{{\left(1-\gamma \right)}^{3}\gamma^{\prime }e^{-2(n+m)/e}}{8{\left(1-\gamma^{\prime }\right)}^{2}}{\left(\frac{M_{1}}{M_{2}}\right)}^{3}\right)}^{K/2}}\\ \; & \leq \left(\kappa +\frac{2N\tau^{q}}{\lambda }\right)\sqrt{\frac{2(\tilde{F}\left(z^{\left(0\right)}\right)-{\tilde{F}}_{⋆})}{\kappa }{\left(1-\frac{C}{{\left(1+2NR^{q}\lambda^{-1}\kappa^{-1}\right)}^{3}}\right)}^{K/2}}, \end{array}$$
where we define $C:=\frac{{\left(1-\gamma \right)}^{3}\gamma^{\prime }e^{-2(n+m)/e}}{8{\left(1-\gamma^{\prime }\right)}^{2}}$ and Lemma 1 is used at the last inequality to replace $M_{1}$, $M_{2}$ and $M_{2}^{\prime }$. We can immediately confirm $C\leq \frac{1}{16}$ from $0<\gamma^{\prime }<\gamma <1$, $\gamma^{\prime }<\frac{1}{2}$, and $e^{-2(n+m)/e}<1$. Finally, by choosing $\kappa =2N\tau^{q}\lambda^{-1}$,
(40)$$\begin{array}{cc} ∥g^{\left(K\right)}{∥}_{2} & \leq \sqrt{\frac{16\left(\tilde{F}\left(z^{\left(0\right)}\right)-{\tilde{F}}_{⋆}\right)N\tau^{q}}{\lambda }{\left(1-\frac{C}{{\left(1+{\left(R/\tau \right)}^{q}\right)}^{3}}\right)}^{K/2}}\\ \; & \leq \sqrt{\frac{16\left(\tilde{F}\left(z^{\left(0\right)}\right)-{\tilde{F}}_{⋆}\right)N\tau^{q}}{\lambda }}r^{K}, \end{array}$$
where we use ${\left(R/\tau \right)}^{q}\geq 1$ (owing to R≥τ by definition) and let $r:={\left(1-C/8\right)}^{1/4}$ and $\sqrt[{4}]{127/128}\leq r<1$. □

**Remark** **2.**
*More precisely, Altschuler et al. [25] showed that the Sinkhorn algorithm converges in $O(N^{2}L^{3}\left(\log N\right)\varepsilon^{-3})$ time, where ${L:=∥D∥}_{\infty }$. For q-DOT, its computational complexity is not directly comparable to that of the Sinkhorn in L; instead, the following analysis provides us a qualitative comparison. First, the convergence rate of q-DOT in Equation (Equation 33) is translated into the iteration complexity $K=O(\log\left(N\varepsilon^{-1}\right)/\log\left(1/r\right))$. The rate r is introduced in the proof of Theorem 1 (see Equation (Equation 40)): $r\geq {\left(1-\frac{C}{{\left(1+{\left(R/\tau \right)}^{q}\right)}^{3}}\right)}^{1/4}$. Then, by the Taylor expansion, we have a rough estimate $K\approx O(N^{2}R^{-3q}\log\left(N\varepsilon^{-1}\right))$, where R is a bound on the possible primal variables defined in Equation (Equation 35). We cannot directly compare $R^{-q}$ and L; nevertheless, $R^{-q}$ and L can be considered in the same magnitude given a reasonably sized domain Z, noting that $\nabla \Omega \left(\pi \right)\approx O\left(\pi^{1-q}\right)$. Hence, it is reasonable to suppose that both the Sinkhorn algorithm and q-DOT roughly converge in cubic time with respect to L.*


## 5. Numerical Experiments

### 5.1. Sparsity

All the simulations described in this section were executed on a 2.7 GHz quad-core Intel^®^ Core^™^ i7 processor. We used the following synthetic dataset: ${\left(x_{i}\right)}_{i=1}^{n}∼N\left(1_{2},I_{2}\right)$, ${\left(y_{j}\right)}_{j=1}^{m}∼N\left(-1_{2},I_{2}\right)$, and n=m=30, where N(μ,Σ) represents the Gaussian distribution with mean μ and covariance Σ. For each of the unregularized OTs, *q*-DOT, and Sinkhorn algorithm, we computed the transport matrices. For *q*-DOT and the Sinkhorn algorithm, different regularization parameters λ were compared: $\lambda \in \left\{1\times 10^{-2},1\times 10^{-1},1\right\}$; and $\varepsilon =1\times 10^{-6}$ was used as the stopping criterion: *q*-DOT stopped after the gradient norm was less than ε, and the Sinkhorn algorithm stopped after the $\ell_{1}$ error of the coupling constraints was less than ε. We compared different deformation parameters $q\in \left\{0,0.25,0.5,0.75\right\}$ and fixed the dual $\ell_{2}$-regularization parameter $\kappa =1\times 10^{-6}$ for *q*-DOT. The *q*-DOT with q=0 corresponded to a regularized OT with the squared 2-norm proposed by Blondel et al. [20]. For the unregularized OT, we used the implementation of the Python optimal transport package [38]. For *q*-DOT, we used the L-BFGS-B method (instead of the vanilla BFGS) provided by the SciPy package [39]. To determine zero entries in the transport matrix, we did not impose any positive threshold to disregard small values (as in Swanson et al. [6]) but regarded entries smaller than machine epsilon as zero.

The simulation results are shown in Table 3 and Figure 3. First, we qualitatively evaluated each method by using Figure 3 such that *q*-DOT obtained a very similar transport matrix to the unregularized OT solution. The solution was slightly blurred with increases in *q* and λ. In contrast, the Sinkhorn algorithm output considerably uncertain transport matrices. Furthermore, the Sinkhorn algorithm was numerically unstable with a very small regularization such as λ=0.01.

From Table 3, we further quantitatively observed the behavior. The transport matrices obtained by *q*-DOT were very sparse in most cases, and the sparsity was close to that of the unregularized OT. Furthermore, we observed the tendency such that smaller *q* and λ yielded a sparser solution. Significantly, the Sinkhorn algorithm obtained completely dense matrices (sparsity = 0). Although the transport matrices of *q*-DOT with (q,λ)=(0.5,1),(0.75,1) appear somewhat similar to the Sinkhorn solutions in Figure 3, the former is much sparser. This suggests that a deformation parameter *q* slightly smaller than 1 is sufficient for *q*-DOT to output a sparse transport matrix.

For the obtained cost values $\langle D,\hat{\Pi }\rangle$, we did not see a clear advantage of using a specific *q* and λ from the results of *q*-DOT. Nevertheless, it is evident that *q*-DOT more accurately estimated the Wasserstein cost than the Sinkhorn algorithm regardless of the *q* and λ used in this simulation.

### 5.2. Runtime Comparison

We compared the runtimes of *q*-DOT and the Sinkhorn algorithm using the same dataset as in Section 5.1, but with different dataset sizes: we chose n=m∈{100,300,500,1000}. The parameter choices were the same as in Section 5.1, except that the regularization parameter was fixed to λ=0.1. The result is shown in Figure 4; the larger deformation parameter *q* makes *q*-DOT converge faster when n=m=100. When n=m≥300, the difference between q=0, q=0.25, and q=0.5 was not as evident. This may be partly because we fixed the parameter choice $\kappa =1\times 10^{-6}$ for the all experiments, unlike the oracle parameter choice $\kappa =2N\tau^{q}\lambda^{-1}$ (in Theorem 1) depending on *q*. Nonetheless, q=0.75 is clearly superior to the smaller *q*. From these observations, the trade-off between the sparsity and computation speed resulting from the deformation parameter *q* is theoretically established in Theorem 1 and it was empirically observed.

### 5.3. Approximation of 1-Wasserstein Distance

Finally, we compared the approximation errors of the 1-Wasserstein distance $|\langle D,\hat{\Pi }\rangle -\langle D,\Pi_{♯}\rangle |$ of *q*-DOT and the Sinkhorn algorithm with different *q* and λ, where $\hat{\Pi }$ represents the computed transport matrix and $\Pi_{♯}\in {arg min}_{\Pi \in U(\mu ,\nu )}\langle D,\Pi \rangle$ represents the LP solution. We used the same dataset and stopping criterion ε as described in Section 5.1 For the range of *q*, we used $q\in \left\{0.00,0.25,0.50,0.75\right\}$. For the range of λ, we used $\lambda \in \left\{0.05,0.1,0.5\right\}$.

The result is shown in Figure 5. The difference was not significant when *q* was small, such as $q\in \left\{0.00,0.25\right\}$. Once *q* became larger, such as $q\in \left\{0.50,0.75\right\}$, the approximation error evidently worsened. The Sinkhorn algorithm always exhibited worse approximation errors than *q*-DOT with *q* in the range used in this simulation regardless of λ. Formal guarantees for the 1-Wasserstein approximation error (such as Altschuler et al. [25] and Weed [40]) will be considered in future work.

## Figures and Tables

**Figure 1 entropy-24-01634-f001:**
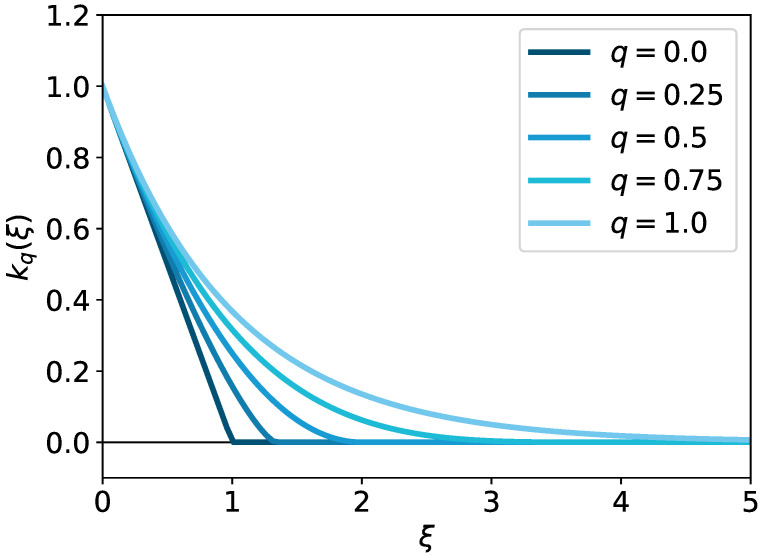
Plots of the *q*-Gibbs kernels with different *q* (λ=1).

**Figure 2 entropy-24-01634-f002:**
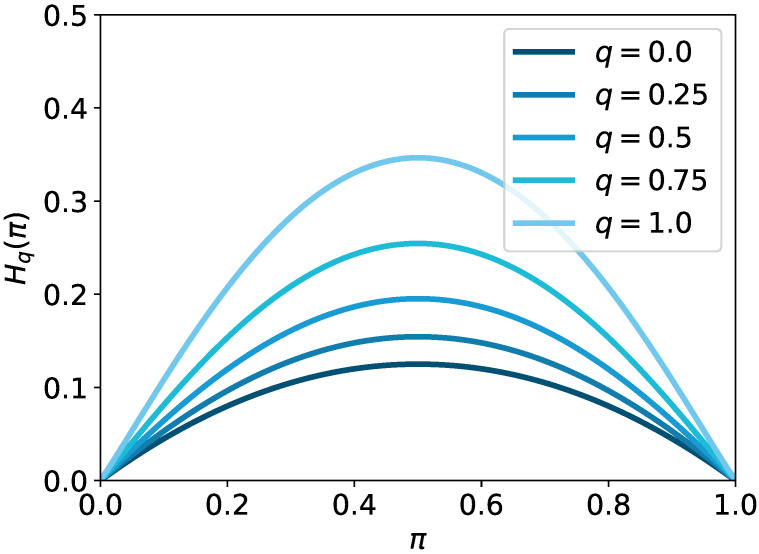
Plots of deformed *q* entropy with different *q* values. A constant term is ignored in the plots so that the end points are calibrated to zero.

**Figure 3 entropy-24-01634-f003:**
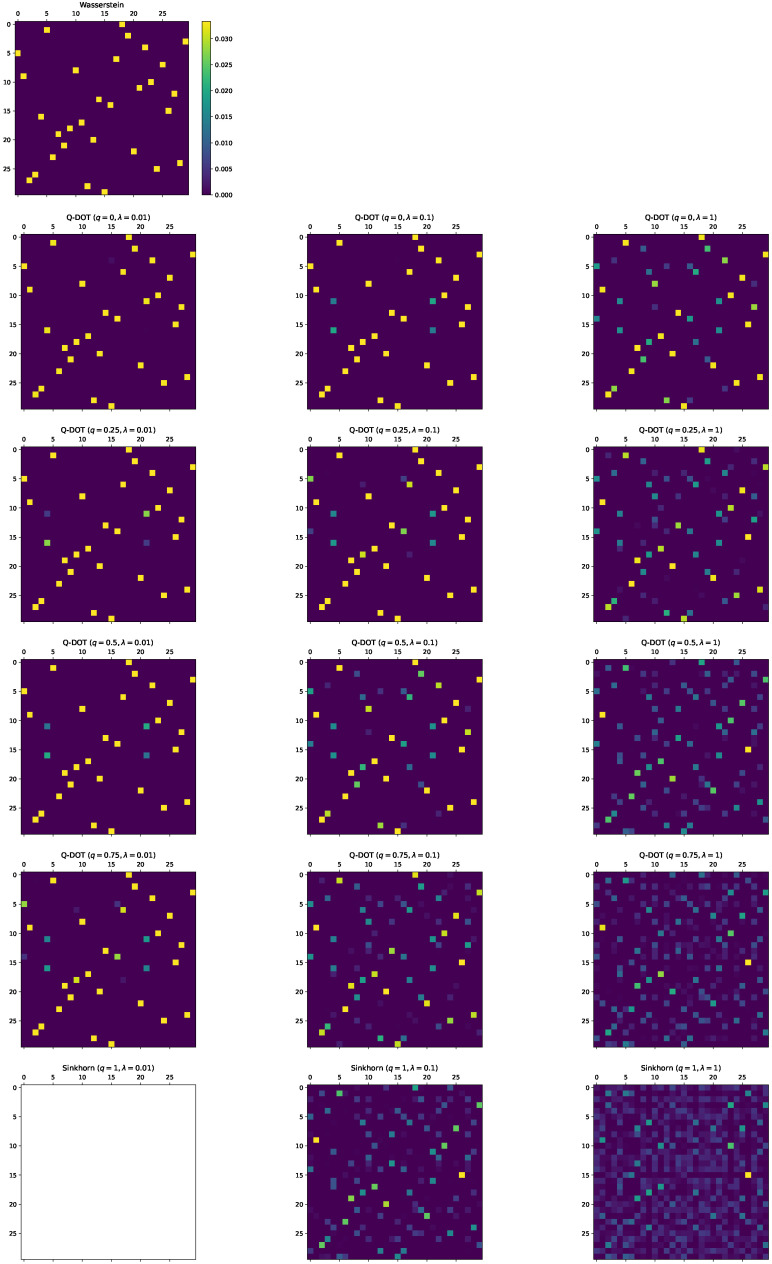
Comparison of transport matrices. Wasserstein represents the result of the unregularized OT. Sinkhorn (λ=0.01) does not work well because of numerical instability.

**Figure 4 entropy-24-01634-f004:**
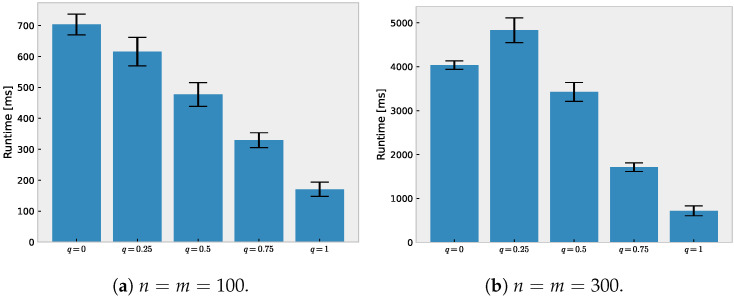
Runtime comparison of *q*-DOT and Sinkhorn algorithm (q=1). The error bars indicate the standard errors of 20 trials.

**Figure 5 entropy-24-01634-f005:**
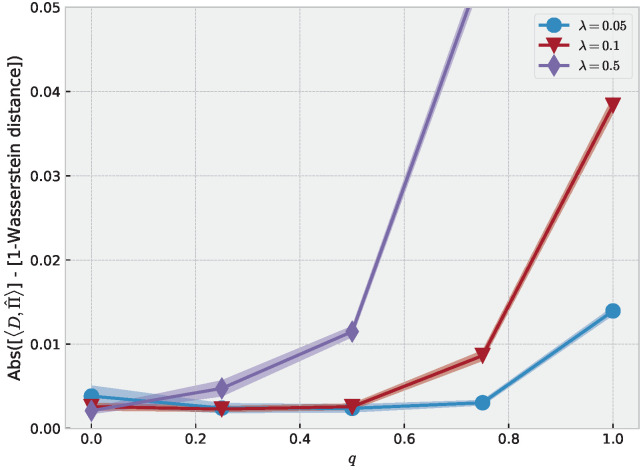
Wasserstein approximation error of *q*-DOT and the Sinkhorn algorithm (q=1). The line shades indicate the standard errors of 20 trials.

**Table 1 entropy-24-01634-t001:** Summary of Ω(π), $\Omega^{⋆}\left(\eta \right)$, and $\nabla \Omega^{⋆}\left(\eta \right)$ for several regularizers. The relationship between Ω, its conjugate, and the derivatives are summarized in Bao and Sugiyama [28].

	Ω(π)	$\Omega^{⋆}\left(\eta \right)$	$\nabla \Omega^{⋆}\left(\eta \right)$
Negative entropy	λ(πlogπ−π)	$\lambda e^{\eta /\lambda }$	$e^{\eta /\lambda }$
Squared 2-norm	$\frac{\lambda }{2}\pi^{2}$	$\frac{1}{2\lambda }{\left[\eta \right]}_{+}^{2}$	$\frac{1}{\lambda }{\left[\eta \right]}_{+}$
Deformed *q* entropy	$\frac{\lambda }{2-q}\left(\pi \log_{q}\left(\pi \right)-\pi \right)$	$\frac{\lambda }{2-q}\exp_{q}{\left(\eta /\lambda \right)}^{2-q}$	$\exp_{q}\left(\eta /\lambda \right)$

**Table 2 entropy-24-01634-t002:** Comparison of the computational complexity of the Sinkhorn algorithm and deformed *q*-optimal transport. $N=\max\left\{n,m\right\}$.

Sinkhorn	*q*-DOT
$O(N^{2}\left(\log N\right)\varepsilon^{-3})$	$O(N^{2}\log\left(N\varepsilon^{-1}\right))$

**Table 3 entropy-24-01634-t003:** Comparison of the sparsity and cost with the synthetic dataset. Sparsity indicates the ratio of zero entries in each transport matrix. We counted the number of entries smaller than machine epsilon to measure the sparsity instead of imposing a small positive threshold for determining zero entries. Sinkhorn (λ=0.01) does not work well because of numerical instability.

	Sparsity	Cost $\langle D,\hat{\Pi }\rangle$
Wasserstein (unregularized)	0.967	7.126
*q*-DOT (q=0.00,λ=0.01)	0.962	7.129
*q*-DOT (q=0.00,λ=0.10)	0.961	7.126
*q*-DOT (q=0.00,λ=1.00)	0.950	7.144
*q*-DOT (q=0.25,λ=0.01)	0.963	7.129
*q*-DOT (q=0.25,λ=0.10)	0.959	7.126
*q*-DOT (q=0.25,λ=1.00)	0.912	7.133
*q*-DOT (q=0.50,λ=0.01)	0.963	7.136
*q*-DOT (q=0.50,λ=0.10)	0.946	7.127
*q*-DOT (q=0.50,λ=1.00)	0.879	7.155
*q*-DOT (q=0.75,λ=0.01)	0.948	7.127
*q*-DOT (q=0.75,λ=0.10)	0.897	7.136
*q*-DOT (q=0.75,λ=1.00)	0.647	7.245
Sinkhorn (λ=0.01)	—	—
Sinkhorn (λ=0.10)	0.000	7.164
Sinkhorn (λ=1.00)	0.000	7.788

## Data Availability

Not applicable.

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
