# Peer review of "Sparse Regularized Optimal Transport with Deformed q-Entropy"

_entropy, 2022, doi:10.3390/e24111634_

Round 1

Reviewer 1 Report

My comments are given in the attached file.

Author Response

We appreciate the reviewer's thoughtful comments on our manuscript. Our point-by-point responses to the comments are as follows:

1. Derivation of q-entropy: We added Appendix A in the revision to describe the step-by-step derivation.

2. Simplification of constants in the proof: In the updated manuscript, it is modified.

3. and 4. The dependence of the computational complexity on the cost matrix size $\|D\|_\infty$: We added Remark 2 in Section 4 to discuss this point. Roughly speaking, both $q$-DOT and the Sinkhorn algorithm depend on the cost matrix size in the cubic order; but, to be precise, they cannot be compared directly because of some differences in the assumptions.

5. Theoretical analysis of the approximation error: The analysis has not been successful so far because q, \lambda, and \kappa (introduced for convergence analysis of BFGS) depend on each other, which makes the analysis involved. We leave it for future work and showed empirical analysis in Section 5 and Appendix D instead.

6. Large-scaled experiments: We added experiments with N=2000 and N=3000 in Appendix D to show the hyperparameter sensitivity of $q$-DOT in response to Reviewer 2 simultaneously. The general trend of sparsity and runtime is similar to the case N≤1000, but we observed a slightly different trend of the absolute error of the computed cost. To investigate the latter trend in detail, we need to deal with approximation error analysis, which is left for future work as mentioned above.

Reviewer 2 Report

The authors propose a new regularization on the optimal transport plan. By controlling the tuning parameter, their regularization term interpolates between negative Shannon entropy and squared L2 norm, where the former is more efficient, but lacks accuracy and the the later is accurate, but expensive in computation.  Overall, I find the results particularly useful and interesting.  Below are some specific comments.

(1) The proposed regularization term looks similar to Tsallis entropy. Could the authors include Tsallis entropy in the simulation study? I think the manuscript will be much stronger if the proposed method demonstrates superior performance over Tsallis entropy.

(2) Could the authors make a unified table that includes both computational time and estimation accuracy for different tuning parameters. This can help the readers find optimal q and lambda more easily. Currently, they have the results in Table 3, Figure 4 and Figure 5 separately.  

(3) I don't quite understand the results in Figure 4. Could the authors explain why runtime has an increase from q = 0 to q=0.25? One reason I can think of is that the number of trials is too small, only 20.

(4) For the simulations in section 5.1, could the authors present results for larger n and m. It's better to match with section 5.2 for comparison purpose.

Author Response

We appreciate the reviewer's thoughtful comments on our manuscript. Our point-by-point responses to the comments are as follows:

1. Comparison with the Tsallis entropy: We added the comparison in Appendix D1, which shows that the Tsallis regularized OT does not behave well in terms of both sparsity and approximation error (at least when applied with the dual optimization by L-BFGS).

2. and 4. Large-scaled datasets with unified tables: In Appendix D2 of the updated manuscript, we added the experimental results with N=2000 and N=3000, and the results are summarized in unified tables for ease of the comparison of hyperparameters. The general trend of sparsity and runtime is similar to the case with N≤1000, while absolute error behaves somewhat differently. The latter trend is a consequence of the difficulty in approximation error analysis, which is beyond the scope of the current work (confer our reply to Reviewer 1 for further details).

3. Runtime increases from q=0 to q=0.25: As we have already discussed in Section 5.2, this may be partly because of the limitation of our analysis (Theorem 1): our convergence analysis requires an oracle parameter choice for \kappa (L2 regularization strength) to get a meaningful trend with respect to N, \lambda, and q. This oracle parameter depends on the unknown parameter \tau (upper bound of transport matrices) and hence may lead to a slightly different trend than we observed in the simulation.

Round 2

Reviewer 1 Report

I thank the authors for their thorough responses to my questions and concerns.